# EviDiff: Learning Object-wise Consistency for Text-to-Image Diffusion

## Abstract

Consistency constraint between the text prompts and the image contents is pivotal in text-to-image (T2I) diffusion models for composing multiple object categories. However, such consistency constraint is often underemphasized in the denoising process of diffusion models. Although token supervised diffusion models can mitigate this issue by learning object-wise consistency between the image content and object segmentation maps, it tends to suffer from the problems of segmentation map bias and semantic overlap conflict, especially when involving multiple objects. To address this, we propose EviDiff, a new evidential learning-supervised T2I diffusion model, which leverages the advantages of uncertainty metric and conflict detection to enhance the fault tolerance of unreliable segmentation maps and suppress semantic conflicts, strengthening object-wise consistency learning. Specifically, a pixel evidence loss is proposed to restrain overconfidence in unreliable labels through evidential regularization, and a token conflict loss is designed to weaken the contradiction between semantics through optimizing a measured conflict factor. Extensive experiments show that our EviDiff outperforms state-of-the-art T2I diffusion models in multi-object compositional generation without requiring additional inference-time manipulations. Notably, our EviDiff can be seamlessly extended to the existing training pipeline of T2I diffusion models. The code and the trained EviDiff model are available at https://github.com/anonymity-coder/EviDiff.

## 1 Introduction

Recent advances in text-to-image (T2I) diffusion models (Chen et al., Feng et al., 2023, Yang et al., 2023, Ramesh et al., 2021, Ramesh et al., 2022, Xue et al., 2023) have demonstrated remarkable progress within computer vision applications, propelling the capabilities of general generative models to a new level. Leveraging their exceptional generation ability, these models enable users to synthesize highly realistic and creative visual content through user-specified text prompts. However, they often struggle to compose multiple objects into a coherent scene (as shown in the SD v1.4 in Fig. 1) since the noise is predicted by a full text prompt in the denoising objective of T2I diffusion models (Wang et al., 2024, Feng et al.), which requires the model to be equipped with the ability of understanding both the full text prompt and individual linguistic concepts from the prompt.

Several studies (Wang et al., 2024, Feng et al., Liu et al., 2022, Liew et al., 2022, Chefer et al., 2023, Ma et al., 2024) have addressed the challenge of generating images including multiple objects. They enable the generated images better reflect the text prompts through utilizing fine-grained text information. As proposed in TokenCompose (Wang et al., 2024), the fundamental idea is to perform consistency constraint between the text prompts and the image contents in the finetuning stage by imposing training objectives at the token level. It adopts Grounding DINO (Liu et al., 2024a) and Segment Anything (SAM) (Kirillov et al., 2023) to obtain the grounding segmentation map of each text token for local supervision of each text token. Once fine-tuned, the model can generate images by composing different combinations of words from text prompts in the inference stage. Although these methods have shown great success, they still suffer from segmentation map bias and semantic overlap conflict, which is problematic particularly when the generated image involves multiple objects (TokenCompose in Fig. 1).

We hypothesize that the limitations come from two aspects. On the one hand, the masks (the segmentation maps) generated directly by SAM are not always reliable (Ji et al., 2024), which results in overfitting to spurious regions. On the other hand, semantic overlaps result in conflicts (Ke et al., 2021) among different text tokens since the consistency constraint between each noun token from the text prompt and its corresponding segmentation map is individually optimized. Therefore, we seek to address the following problems: *How to cope with segmentation map bias and semantic overlap conflict for more effective consistency constraint in T2I diffusion models?*

*Motivation:* Inspired by the above observation, we find it desirable to tackle such problems by considering the uncertainty metric of pixel-level predictions and quantifying the semantic overlap conflict. Evidential Deep Learning

Figure 1: Our proposed EviDiff significantly enhances the baseline model on text condition following, demonstrating superior capability in composing multiple object categories.

(EDL) (Sensoy et al., 2018, Malinin & Gales, 2018), which is able to collect evidence of each category and estimate the epistemic uncertainty in a single forward pass. Given this, we consider each pixel of each noun token from the text prompt of as a classification unit and provide evidence values per class to reflect both the confidence and the associated uncertainty. In this way, the model tends to produce low-confidence predictions in mislabeled regions of the biased segmentation map, thereby avoiding overfitting to incorrect supervision. Moreover, Dempster-Shafer Evidence Theory (DST) (Dempster, 2008), an evidence combination rule, allows beliefs from different evidences to be combined to obtain a new belief (Sentz & Ferson, 2002, Jøsang & Hankin, 2012). It is able to quantify the conflict level through introducing the conflict penalty item in overlapping areas. Inspired by this, we consider each noun token from the text prompt as different evidences and utilize DST combination rule to measure their conflict. Therefore, the model can perceive overlapping conflicts between different semantic evidences, effectively alleviating the semantic competition problem caused by independent optimization.

To this end, we propose EviDiff, an end-to-end fine-tuning strategy to enhance multi-object composition through a novel consistency constraint (i.e., EDL driven object-wise consistency constraint) between the text prompts and the image contents. Specifically, a pixel evidence loss is proposed to cope with the segmentation map bias. It models the per-token cross-attention maps as evidence distributions (i.e., Dirichlet distribution) and calculates the cross entropy using the mean value of the predicted Dirichlet distribution instead of directly using point predictions. In this way, when the uncertainty is high (i.e., the distribution is loose), the loss of EDL will not excessively penalize the model, allowing the model to remain cautious when facing biased segmentation map and avoid being forced to learn incorrectly. In addition, a token conflict loss is proposed to alleviate semantic overlap conflict. It regards per-token cross-attention map as multiple evidence embeddings and utilizes DST combination rule to aggregate such multiple evidence embeddings for conflict measurement. The measured conflict is utilized as an optimization target, enabling the model to adaptively adjust the spatial distribution of different tokens' cross-attention maps, thus mitigating semantic conflicts arising from attention map overlap.

Our main contributions are summarized as the following:

- We formulate the problems of segmentation bias and semantic overlap conflict during the object-wise consistency constraint between the text prompts and the image contents. And we propose a new T2I diffusion model, EviDiff, which pioneeringly introducing EDL based pixel-level uncertainty metric and DST based token-level conflict quantification to address these two problems. The proposed EviDiff can be seamlessly integrated into the existing training pipeline of the T2I diffusion model.

- We design a pixel evidence loss to restrain overconfidence in unreliable labels through simultaneously considering pixel-level classification prediction and uncertainty estimation.

- We propose a token conflict loss to weaken the semantic overlap conflict through treating the conflict factor measured by the DST combination rule as the optimization objective.

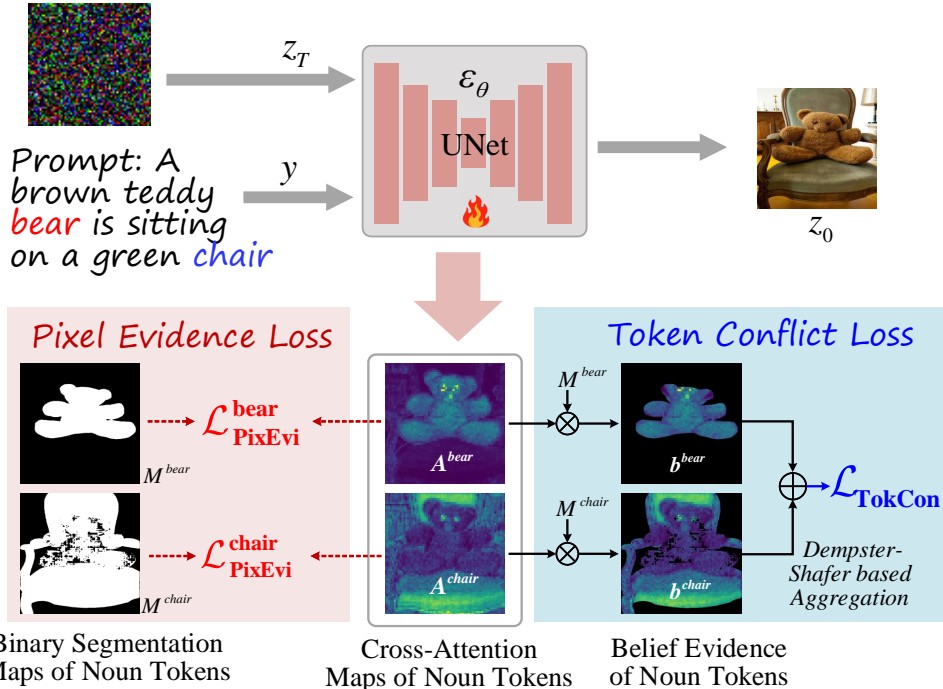

Figure 2: **Overview of EviDiff.** The main components include: (1) The pixel evidence loss drives reliable consistency constraint between the text prompts and the image contents through pixel-level uncertainty estimation; (2) The token conflict loss weakens the contradiction between semantics.

- Extensive comparisons with previous outstanding methods demonstrate that EviDiff effectively improves the performance in composing multiple objects without additional inference cost.

## 2 METHOD

Our EviDiff (Fig. 2) proposes a pixel evidence loss to restrain overconfidence in unreliable labels when conducting pixel-level supervision between the image content and the object segmentation map, and designs a token conflict loss to weaken the contradiction between semantics through optimizing a measured conflict factor. In Section 2.2, we first illustrate the problem setup of the segmentation map bias. Following this, we detail how the pixel evidence loss address this through pixel-level uncertainty metric. Subsequently, in Section 2.3, we clarify the semantic overlap conflict and demonstrate how the token conflict loss alleviates such conflict through DST combination rule.

### 2.1 PROBLEM SETUP

**Segmentation map bias.** While consistency constraint based T2I diffusion methods have shown great success (Wang et al., 2024, Zhou et al., 2025), the generation effect heavily depends on the quality of the segmentation map used for consistency constraint. For example, when performing a training constraint between the text token guided cross-attention map and the corresponding binary segmentation map, the model overfits to incorrect regions if binary segmentation map is biased. We mathematically formulate this problem in Appendix B.1.

**Semantic overlap conflict.** Although $\mathcal{L}_{pixel}$ can focus the cross-attention map of nouns in text prompts on the target area, a side effect of this loss is that separately optimizing each cross-attention map is contradictory when different semantics overlap (Kim et al., 2025). To be specific, this per-noun supervision strategy inherently assumes that the visual concepts of different nouns are spatially disjoint. For example, in the phrase "a cat on a chair," the noun "cat" and "chair" may correspond to overlapping regions in the image due to physical interaction or spatial proximity. Supervising each noun's cross-attention map with an independent segmentation map leads to semantic overlap conflicts, where the same pixel may be simultaneously assigned to multiple concepts. This introduces contradicting gradients during training, impairing the model's ability of concept-to-region mappings. We mathematically formulate this problem in Appendix B.2.

Figure 3: **Detailed diagram of pixel evidence loss and token conflict loss.** For pixel evidence loss, the noun token's cross-attention maps are fed into the evidential Net to generate noun token's evidences. Then, these evidences are used to build the Dirichlet distribution, outputting pixel-level probabilities and the uncertainty for reliable pixel-level supervision. For token conflict loss, the conflict coefficient of the overlapping area between noun token's cross-attention maps is calculated for optimization. Moreover, the activations of different noun tokens are aggregated into respective corresponding spatial regions.

## 2.2 PIXEL EVIDENCE LOSS

To address the segmentation bias, our key insight is to add uncertainty metric between the cross-attention map and the segmentation map to achieve more reliable consistency constraint. We accomplish this by leveraging the epistemic uncertainty ability of EDL, which can quantify the credibility of its predictions. Instead of treating the attention score of each pixel as a deterministic prediction, we reinterpret it as evidence supporting its foreground or background class. As shown in Fig. 3, these evidence values are modeled via a Dirichlet distribution to construct an uncertainty-aware attention learning process.

*Evidence-based Attention Modeling.* Given a cross-attention map $A$ and the corresponding binary segmentation map $M$, $A$ is fed into an MLP evidence network $f_\phi(\cdot)$ with an activation function (Softplus) to map $A$ into the evidence space, thus outputting evidence values $e$. For each pixel $A_{i,j}$, its evidence is defined as $e_{i,j} \in \mathbb{R}_{\geq 0}$, from which the parameters of a Dirichlet distribution $D(p \mid \alpha)$ are obtained as $\alpha_{i,j} = e_{i,j} + 1$. The Dirichlet distribution $D(p \mid \alpha)$ is considered as the conjugate prior to the multinomial distribution. It provides a predictive distribution for the segmentation results, defned as follows:

$$D(p \mid \alpha) = \begin{cases} \frac{1}{\beta(\alpha)} \prod_{k=1}^{K} p_k^{\alpha_k - 1} & for \quad p \in \Omega \\ 0 & otherwise \end{cases} \quad (1)$$

where $p = [p_1, \ldots p_k]$ are the parameters of the multinomial distribution, $\beta(\alpha)$ is the high-dimensional multinomial beta function, and $\Omega$ is the $K - 1$ dimensional unit simplex, defned as:

$$\Omega = \left\{ p \left| \sum_{c=1}^{C} p^c = 1 \text{ and } 0 \leq p^1, \ldots, p^C \leq 1 \right. \right\} \quad (2)$$

Subsequently, Subjective Logic (Jsang, 2018) is applied for the optimization of Dirichlet distribution parameter optimization. It establishes a theoretical foundation linking Dirichlet distribution parameters to confidence and uncertainty

quantification. Specifcally, for the predicted cross-attention map $A$, it provides a mass belief and uncertainty, satisfying:

$$u_{i,j}^c + \sum_{c=1}^{C} b_{i,j}^c = 1 \tag{3}$$

where $b_{i,j}$ and $u_{i,j}^c$ are the cross-attention map probability of the pixel $(i,j)$ for the $c$-th class and the uncertainty of the pixel $(i,j)$, respectively. The the mass of belief and uncertainty for the pixel $(i,j)$ can be expressed as follows:

$$b_{i,j}^c = \frac{e_{i,j}^c}{S} = \frac{\alpha_{i,j}^c - 1}{S} \quad \text{and} \quad u_{i,j} = \frac{C}{S} \tag{4}$$

where $S = \sum_{c=1}^{C} \left(e_{i,j}^c + 1\right)$ is the Dirichlet strength. It describes that the higher the allocated belief mass, the more evidence is obtained for pixel $(i,j)$. Conversely, the less evidence obtained, the greater the overall uncertainty for the segmentation of pixel $(i,j)$.

Considering that, on the simplex, the ideal Dirichlet distribution should concentrate its mass on the vertex corresponding to the true class label. The distribution parameters should be close to 1 for all incorrect classes and significantly larger for the correct one. Therefore, it is necessary to design a loss function that guides the model to optimize the Dirichlet parameters in a way that minimizes segmentation uncertainty. Specifcally, We use the cross-entropy loss $\mathcal{L}_{ce} = \sum_{c=1}^{C} -y^c \log\left(p^c\right)$ to link the Dirichlet distribution with the belief representation in subjective logic. Based on the evidence, this loss can be reformulated to reflect the expected prediction under the Dirichlet distribution, enabling the model to learn both accurate and uncertainty-aware segmentations, defned as follows:

$$\mathcal{L}_{ice} = \int \left[ \sum_{c=1}^{C} -y^c \log(p^c) \right] \frac{1}{\beta(\alpha)} \prod_{c=1}^{C} (p^c)^{\alpha^c - 1} \, dp = \sum_{c=1}^{C} y^c \left(\psi\left(S\right) - \psi\left(\alpha^c\right)\right) \tag{5}$$

where $y^c$ is the ground truth labels and $\psi\left(\cdot\right)$ is digamma function. We further incorporate a Kullback-Leibler (KL) divergence loss to suppress evidence for incorrect classes while preventing the Dirichlet parameter of the ground-truth class from being forced to 1, defined as:

$$\mathcal{L}_{KL} = \log \left( \frac{\Gamma\left(\sum_{c=1}^{C} \tilde{\alpha}^c\right)}{\Gamma(C) \sum_{c=1}^{C} \Gamma\left(\tilde{\alpha}^c\right)} \right) + \sum_{c=1}^{C} \left(\tilde{\alpha}^c - 1\right) \left[ \psi\left(\tilde{\alpha}^c\right) - \psi\left(\sum_{c=1}^{C} \tilde{\alpha}^c\right) \right] \tag{6}$$

where $\tilde{\boldsymbol{\alpha}}^c = y^c + (1 - y^c) \odot \boldsymbol{\alpha}^c$ and $\Gamma\left(\cdot\right)$ is the gamma function.

The loss $\mathcal{L}_{\text{PixEvi}}$ in this section consists of cross-entropy loss $\mathcal{L}_{ce}$, $\mathcal{L}_{ice}$, and $\mathcal{L}_{KL}$.

$$\mathcal{L}_{\text{PixEvi}} = \frac{1}{N} \sum_{n=1}^{N} (\lambda_1 \mathcal{L}_{ce}^{(n)} + \lambda_2 (\mathcal{L}_{ice}^{(n)} + \mathcal{L}_{KL}^{(n)})) \tag{7}$$

where $n$ represents each text token that belongs to a noun within the text prompt, and $\lambda_1$, $\lambda_2$, and $\lambda_3$ are hyperparameters to balance these three losses. The cross-entropy loss $\mathcal{L}_{ce}$ is adopted to maximize the consistency between the cross-attention map and the binary segmentation map.

## 2.3 TOKEN CONFLICT LOSS

To solve the semantic overlap conflict, we propose to aggregate all noun-level cross-attention maps using Dempster-Shafer Evidence Theory (DST) (Dempster, 2008) to explicitly quantify inter-noun conflicts (Fig. 3), guiding the model to learn semantically decoupled attention regions.

*Cross-Attention Map Aggregation.* Given a set of per-noun cross-attention maps $\{A^1, A^2, \ldots, A^N\}$, we construct basic belief assignment function based on DST.

$$b^n = \gamma^n A^n \cdot M^n \tag{8}$$

where $b^n$ is the belief evidence belonging to the $n$-th noun category. $\gamma^n \in (0,1)$ represents learnable factor. $M^n$ is the corresponding binary segmentation map. For tow nouns $(v, w)$, the conflict coefficient within the overlapping area $M^v \cap M^w$ is calculated by:

$$K_{i,j}^{v,w} = \begin{cases} b_{i,j}^v \cdot b_{i,j}^w, & \text{if}(i,j) \in M^v \cap M^w \\ 0, & \text{otherwise} \end{cases} \tag{9}$$

Subsequently, we define a intra-object consistency loss $\mathcal{L}_{\text{intra}}$ to aggregate its activations of the cross-attention map into certain subregions of its target regions and a inter-object conflict loss $\mathcal{L}_{\text{inter}}$ to punish conflicting activations in overlapping areas, defined as:

$$\mathcal{L}_{\text{intra}} = \sum_n (1 - b^n)^2 \tag{10}$$

$$\mathcal{L}_{\text{inter}} = \sum_{(i,j) \in M^v \cap M^w} \frac{K_{i,j}^{v,w}}{1 - K_{i,j}^{v,w}} \cdot \sigma(K_{i,j}^{v,w} - \tau) \tag{11}$$

where $\sigma$ is Sigmoid function and $\tau$ is constraint threshold. The final token conflict loss is the sum of $\mathcal{L}_{\text{intra}}$ and $\mathcal{L}_{\text{inter}}$, defines as:

$$\mathcal{L}_{\text{TokCon}} = \frac{1}{N} \sum_{n=1}^{N} (\eta_1 \mathcal{L}_{\text{intra}}^{(n)} + \eta_2 \mathcal{L}_{\text{inter}}^{(n)}) \tag{12}$$

Finally, our EviDiff is jointly optimized by $\mathcal{L}_{LDM}$, $\mathcal{L}_{\text{EviDiff}}$, and $\mathcal{L}_{\text{TokCon}}$. The training objective is $\mathcal{L}_{\text{EviDiff}} = \mathcal{L}_{LDM} + \mathcal{L}_{\text{PixEvi}} + \mathcal{L}_{\text{TokCon}}$

## 3 EXPERIMENTS

### 3.1 DATASETS AND EVALUATION

**Dataset for Finetuning.** We follow the dataset setting in TokenCompose (Wang et al., 2024), which is a subset of COCO image-caption pairs (Lin et al., 2014). To be specific, all unique images are from the Visual Spatial Reasoning dataset (Liu et al., 2023) since these image-caption pairs have relatively low ambiguity in its visual language and the rich diversity of object categories. The CLIP model (Radford et al., 2021) is used to choose the caption that has the highest semantic correspondence to its paired image. The Grounded-SAM (Kirillov et al., 2023, Liu et al., 2024b) is utilized to generate binary segmentation maps of all nouns from the captions. Finally, 4526 image-caption pairs and their corresponding binary segmentation maps are finally obtained.

**Evaluation metrics.** We measure the following metrics: VISOR (Gokhale et al., 2022), MULTIGEN (Wang et al., 2024), Fréchet Inception Distance (FID) (Heusel et al., 2017), CLIP Score (Kumari et al., 2023), Efficiency, and T2I-CompBench (Huang et al., 2023). The detailed evaluation dataset and calculation rules based on these metrics are described in Appendix C.

### 3.2 IMPLEMENTATION DETAILS AND BASELINE METHODS

**Implementation Details.** We follow the experiment settings outlined in TokenCompose (Wang et al., 2024), fine-tuning our EviDiff based on Stable Diffusion v1.4 (Rombach et al., 2022). All experiments are implemented with PyTorch and carried out with NVIDIA GeForce RTX 3090 GPU. The number of training steps is 24000, the initial learning rate is 5e-6 with AdamW (Loshchilov & Hutter), and the batch size is 1 and 4 gradient accumulation steps on a single GPU. Apart from the original denoised target $\mathcal{L}_{LDM}$, $\mathcal{L}_{PixEvi}$ and $\mathcal{L}_{\text{PixEvi}}$ and $\mathcal{L}_{\text{TokCon}}$ are utilized in the finetuning process, where $(\lambda_1, \lambda_2, \lambda_3)$ is set to (5e-5, 5e-7, 5e-7·$min\{1, n_{epoch}/100\}$ for $\mathcal{L}_{\text{PixEvi}}$ and $(\eta_1, \eta_2)$ is set to (1e-4, 1e-4) for $\mathcal{L}_{\text{TokCon}}$. Appendix E provides the pseudocode of finetuning and inferencing.

**Baseline Methods.** To validate the effectiveness of our EviDiff, we compare our method with several representative T2I generation methods, such as Stable Diffusion (Rombach et al., 2022), Composable Diffusion (Liu et al., 2022), Layout Guidance Diffusion (Chen et al., 2024), Structured Diffusion (Feng et al.), Attend-and-Excite (Chefer et al., 2023), SD3 (Esser et al., 2024), and TokenCompose (Wang et al., 2024). See Appendix D for details of these baselines.

### 3.3 MAIN RESULTS

**Results of Multi-category Instance Composition: Object Accuracy (OA) & MULTIGEN**    We quantitatively evaluate the compositionality of EviDiff based on OA (a metric from VISOR) (Gokhale et al., 2022) and MULTIGEN (Wang et al., 2024) compared to the outstanding T2I models. As demonstrated in Table 1, EviDiff achieves state-of-the-art performance on OA, which is 0.0348 higher than the suboptimal TokenCompose (Wang et al., 2024). For MG2-5, it is clear that our EviDiff and TokenCompose show significant improvements apart from MG5. To verify whether our method is benefit for finetuning other versions of T2I model, we add our fine-tuning strategy to Stable Diffusion v2.1 and SD v3 Medium. As shown in Table 2, the results of Multi-category Instance Composition and Photorealism indicate that our EviDiff significantly improves Stable Diffusion v2.1 and noticeably outperforms To-kenCompose. On SD v2.1, compared to TokenCompose, our fine-tuning method improved OA by 0.1106 and MG4 by

Table 1: Evaluation results of our model against baselines. EviDiff holistically demonstrates the best performance regarding Multi-category Instance Composition, Photorealism, Realism and Efficiency. The best score is in blue , with the second-best score in green. (C) denotes the COCO instance validation set and (F) denotes the Flickr30K instance validation set.

| Model | Multi-category Instance Composition | | | | | Photorealism | | | | Efficiency |
|---|---|---|---|---|---|---|---|---|---|---|
| | OA↑ | MG2↑ | MG3↑ | MG4↑ | MG5↑ | FID(C)↓ | FID(F)↓ | CLIP(C)↑ | CLIP(F)↑ | Latency↓ |
| SD v1.4 (Jsang, 2018) | 0.2986 | $0.9072_{1.33}$ | $0.5074_{0.89}$ | $0.1148_{0.45}$ | $0.0088_{0.21}$ | 22.06 | 57.24 | 0.3022 | 0.3082 | $7.54_{0.17}$ |
| Composable (Liu et al., 2022) | 0.2783 | $0.6333_{0.59}$ | $0.2187_{1.01}$ | $0.0325_{0.45}$ | $0.0023_{0.18}$ | 21.71 | 60.32 | 0.3123 | 0.2987 | $13.81_{0.15}$ |
| Layout (Chen et al., 2024) | 0.4359 | $0.9322_{0.69}$ | $0.6015_{1.58}$ | $0.1949_{0.88}$ | $0.0227_{0.44}$ | 23.18 | 58.77 | 0.2851 | 0.3042 | $18.89_{0.20}$ |
| Structured (Feng et al.) | 0.2964 | $0.9040_{1.06}$ | $0.4864_{1.32}$ | $0.1071_{0.92}$ | $0.0068_{0.25}$ | 22.09 | 56.73 | 0.2706 | 0.2931 | $7.74_{0.17}$ |
| Attend-Excite (Chefer et al., 2023) | 0.4513 | $0.9364_{0.76}$ | $0.6510_{1.24}$ | $0.2801_{0.90}$ | $0.0601_{0.61}$ | 21.57 | 57.05 | 0.3018 | 0.3044 | $25.43_{4.89}$ |
| TokenCompose (Wang et al., 2024) | 0.5215 | $0.9808_{0.40}$ | $0.7616_{1.04}$ | $0.2881_{0.95}$ | $0.0328_{0.48}$ | 21.12 | 56.93 | 0.3112 | 0.3205 | $7.56_{0.14}$ |
| **EviDiff(Ours)** | 0.5563 | $0.9736_{0.65}$ | $0.7660_{1.15}$ | $0.2962_{1.58}$ | $0.0353_{0.47}$ | 20.84 | 55.36 | 0.3204 | 0.3299 | $7.54_{0.24}$ |

Table 2: To evaluate the model generalization, we apply our finetuning strategy to Stable Diffusion v2.1 and SD v3 Medium, and report results on multi-category instance composition, FID, and CLIP score.

| Model | Multi-category Instance Composition | | | | Photorealism | | | |
|---|---|---|---|---|---|---|---|---|
| | OA↑ | MG3↑ | MG4↑ | MG5↑ | FID(C)↓ | FID(F)↓ | CLIP(C)↑ | CLIP(F)↑ |
| SD v2.1 frozen | 0.4728 | 0.7014 | 0.2557 | 0.0327 | 21.79 | 56.83 | 0.3045 | 0.3159 |
| SD v2.1 ft. w. $\mathcal{L}_{LDM}$ | 0.5509 | 0.7643 | 0.3207 | 0.0473 | 21.94 | 57.02 | 0.3122 | 0.3231 |
| SD v2.1 ft. w. TokenCompose | 0.6010 | 0.8048 | 0.3669 | 0.0571 | 20.77 | 56.43 | 0.3209 | 0.3264 |
| SD v2.1 ft. w. Ours | 0.7116 | 0.8870 | 0.5401 | 0.1392 | 20.09 | 55.12 | 0.3241 | 0.3348 |
| SD v3 frozen | 0.8084 | 0.9996 | 0.9696 | 0.7328 | 18.21 | 54.33 | 0.3185 | 32.66 |
| SD v3 ft. w. $\mathcal{L}_{LDM}$ | 0.8010 | 0.9996 | 0.9702 | 0.7345 | 18.94 | 55.14 | 0.3174 | 32.43 |
| SD v3 ft. w. TokenCompose | 0.8233 | 0.9997 | 0.9839 | 0.7569 | 18.05 | 54.87 | 0.3198 | 32.87 |
| SD v3 ft. w. Ours | 0.8644 | 0.9997 | 0.9964 | 0.8171 | 17.66 | 54.79 | 0.3247 | 32.78 |

0.1732. In addition, on SD v3 Medium, our method has also achieves competitive performance. These improvements are largely attributed to the consistency constraint between the text prompts and the image contents, which greatly enhances the model's ability of composing multiple object categories. By finetuning the Stable Diffusion thorugh pixel evidence loss and token conflict loss, EviDiff provides reliable consistency constraint during the denoising process and achieves outstanding generation results involving multiple objects. For qualitative experiments, as shown in Fig. 5, EviDiff achieves a high level of image quality in conjunction with multi-category instance composition, compared with outstanding T2I models. The generation results of different finetuning methods based on Stable Diffusion v2.1 are provided in Appendix G.

**Results of Photorealism: FID & CLIP Score**     As shown in Table 1, our EviDiff consistently outperforms outstanding T2I models in both FID (Heusel et al., 2017) and CLIP Score (Kumari et al., 2023). Specifically, EviDiff is 0.28 and 1.37 lower than the suboptimal method on Fid(C) and Fid(F), respectively. It also 0.0081 and 0.0094 higher than the suboptimal method on CLIP Score(C) and CLIP Score(F) respectively. We attribute these performance advantages to the proposed pixel evidence loss and token conflict loss, which enhances image photorealism when composing multiple categories of instances.

**Results of Efficiency: Latency**     Compared to the standard T2I diffusion, our EviDiff does not require additional inference time during the inference stage. As shown in Table 1, EviDiff achieves the best generation performance using the least amount of inference time. The latency results in Table 1 represent the number of seconds required to generate an image with 50 DDIM steps.

**Results of Attribute Binding and Object Relationship: T2I-CompBench**     As illustrated in the Table 3, our EviDiff observes significant gains in all seven evaluation tasks. compared with other finetuing methods. It is clear that EviDiff show significant improvements in color and spatial compared with the TokenCompose finetuning method. It is notable that Evidiff improves attribute binding and object relationship by enhancing the model's multi-object composition ability without specifically optimizing color, shape, texture, and the relationships between objects.

### 3.4 ABLATION STUDY

**Importance of Pixel Evidence Loss** $\mathcal{L}_{\text{PixEvi}}$     As shown in Table 4, we show the Multi-category Instance Composition and Photorealism results, aiming to identify the effectiveness of pixel evidence loss. It is clear that without the

Table 3: Evaluation results about compositionality on T2I-CompBench. Based on SD v2.1 and SD v3 Medium, EviDiff consistently shows the best performance regarding attribute binding, object relationships, numeracy and complex.

| Model | Attribute Binding | | | Object Relationship | | Numeracy↑ |
|---|---|---|---|---|---|---|
| | Color↑ | Shape↑ | Texture↑ | Spatial↑ | Non-Spatial↑ | |
| SD v2.1 frozen | 0.5309 | 0.4447 | 0.4929 | 0.1355 | 0.3099 | 0.4896 |
| SD v2.1 ft. w. $\mathcal{L}_{LDM}$ | 0.5514 | 0.5007 | 0.5376 | 0.1644 | 0.3192 | 0.5042 |
| SD v2.1 ft. w. TokenCompose | 0.6063 | 0.4934 | 0.6131 | 0.1789 | 0.3185 | 0.5094 |
| SD v2.1 ft. w. Ours | 0.6604 | 0.5125 | 0.6287 | 0.2238 | 0.3204 | 0.5366 |
| SD v3 frozen | 0.8033 | 0.5831 | 0.7154 | 0.3102 | 0.4010 | 0.6043 |
| SD v3 ft. w. $\mathcal{L}_{LDM}$ | 0.8152 | 0.5641 | 0.7184 | 0.3361 | 0.4122 | 0.5961 |
| SD v3 ft. w. TokenCompose | 0.8314 | 0.5766 | 0.7463 | 0.3651 | 0.4574 | 0.6243 |
| SD v3 ft. w. Ours | 0.8466 | 0.6037 | 0.7355 | 0.3984 | 0.4722 | 0.6520 |

Table 4: Ablation studies of different objectives with $\mathcal{L}_{LDM}$, $\mathcal{L}_{\text{PixEvi}}$, and $\mathcal{L}_{\text{TokCon}}$.

| Model | $\mathcal{L}_{LDM}$ | $\mathcal{L}_{\text{PixEvi}}$ | $\mathcal{L}_{\text{TokCon}}$ | Multi-category Instance Composition | | | | | Photorealism | | | |
|---|---|---|---|---|---|---|---|---|---|---|---|---|
| | | | | OA↑ | MG2↑ | MG3↑ | MG4↑ | MG5↑ | FID(C)↓ | FID(F)↓ | CLIP(C)↑ | CLIP(F)↑ |
| SD v1.4 | | | | 0.2986 | $0.9072_{1.33}$ | $0.5074_{0.89}$ | $0.1148_{0.45}$ | $0.0088_{0.21}$ | 22.06 | 57.24 | 0.3022 | 0.3082 |
| SD v1.4 | ✓ | | | 0.3821 | $0.9144_{0.21}$ | $0.6372_{1.11}$ | $0.1956_{0.94}$ | $0.1897_{0.32}$ | 24.57 | 60.14 | 0.3028 | 0.3106 |
| SD v1.4 | ✓ | ✓ | | 0.5466 | $0.9821_{0.30}$ | $0.7648_{0.88}$ | $0.2934_{1.14}$ | $0.3361_{0.21}$ | 21.17 | 55.98 | 0.3178 | 0.3246 |
| SD v1.4 | ✓ | ✓ | ✓ | 0.5563 | $0.9736_{0.65}$ | $0.7660_{1.15}$ | $0.2962_{1.58}$ | $0.3530_{0.47}$ | 20.84 | 55.36 | 0.3204 | 0.3299 |

use of the pixel evidence loss, Multi-category Instance Composition and Photorealism have significantly worsened. This is because the text prompt and the object in the image are not aligned during the denoising process, leading to poor multi-categories generation.

**Effectiveness of Token Conflict Loss** $\mathcal{L}_{\text{TokCon}}$    We demonstrate the advantages of token conflict loss by introducing the token conflict loss into the pixel evidence loss. As shown in Table 4, compared to before adding the token conflict loss, the performance metrics of Multi-category Instance Composition and Photorealism after adding the token conflict loss achieves consistent improvements apart from MG2. This is because the token conflict loss avoids multi-object semantic overlap conflicts in the image when conducting the consistency constraint between text prompts and image contents.

The ablation visualization (Fig. 4) clearly shows that fine-tuning Stable Diffusion with only $\mathcal{L}LDM$ struggles to accurately capture the target objects. When the pixel evidence loss $\mathcal{L}PixEvi$ is incorporated, the model improves the reliability of object localization. Furthermore, adding the token conflict loss $\mathcal{L}_{TokCon}$ results in a enhancement of text-to-object consistency. The ablation visualization demonstrates the effectiveness of the proposed loss in capturing the objects corresponding to nouns in the text.

## 4    RELATED WORK

**Text-to-Image Synthesis.** The field of text-to-image synthesis (Du et al., 2023, Hao et al., 2023, Podell et al., Sun et al., 2023, Xu et al., 2024, Zhang et al., 2023) has demonstrated remarkable generative power in various fields, such as text-to-image generation (Rombach et al., 2022, Saharia et al., 2022), text-to-video generation (Blattmann et al., 2023, Singer et al.), and text-to-3D generation (Poole et al., Lin et al., 2023). By encoding text prompts into a condition vector using a pre-trained CLIP (Radford et al., 2021), T2I diffusion models have achieved successful applications in image generation. Although progress has been made, compositional generation still

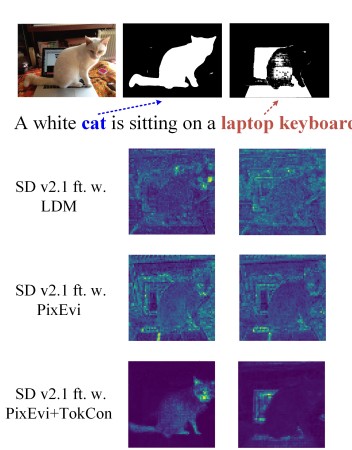

A white **cat** is sitting on a **laptop keyboard**.

SD v2.1 ft. w. LDM

SD v2.1 ft. w. PixEvi

SD v2.1 ft. w. PixEvi+TokCon

Figure 4: Ablation visualization demonstrates that finetuning the Stable Diffusion with only $\mathcal{L}_{LDM}$ does not clearly identify the target objects. By introducing $\mathcal{L}_{PixEvi}$ and $\mathcal{L}_{TokCon}$, the model shows substantial improvement in text-object correspondence.

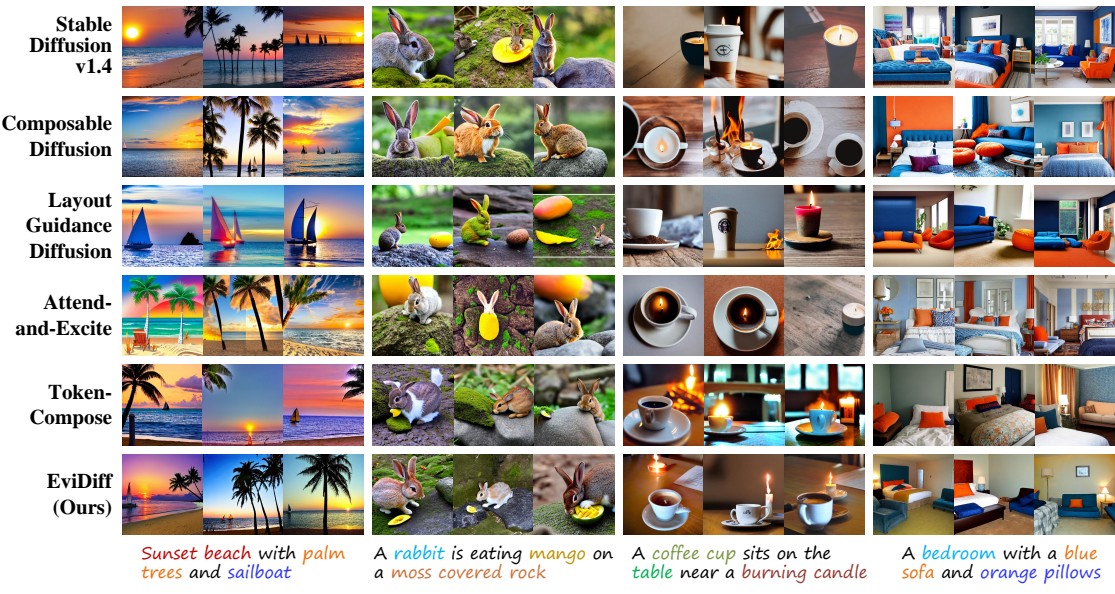

Figure 5: Qualitative comparison between our EviDiff and the other T2I models.

struggles when text prompts involve multiple objects and intri-
cate relationships (Wu et al., 2024). One potential method to generate multi-target images is manipulating the latent and/or cross-attention maps (Liu et al., 2022, Chen et al., 2024, Chefer et al., 2023, Feng et al., Rassin et al., 2023, Wang et al., 2024, Jsang, 2018, Ma et al., 2024). More recently, TokenCompose (Wang et al., 2024) optimizes the cross-attention maps based on segmentation maps to provide dense consistency constraint. It achieves strong compositionality and image quality without additional inference time. Despite its superiority, it suffers from the segmentation bias and optimizing multiple objects individually leads to semantic overlap conflicts, resulting in degraded image generation quality. In this work, we present a different approach to address these problems for more reliable multi-category image composition through consistency constraint with evidential supervision.

**Evidential Deep Learning.** EDL is gradually developed and refined based on Dempster-Shafer theory of evidence (Dempster, 2008) and Subjective Logic theory (Bao et al., 2021). The core idea of EDL is to collect evidence of each category and construct a Dirichlet distribution parameterized over the collected evidence to model the distribution of class probabilities. In addition to the probability of each category, the predictive uncertainty can be quantified from the distribution by Subjective Logic theory in the forward pass. EDL has been applied in a variety of research areas, e.g., EDL based classification (Fu et al., 2023, Fu et al., 2023, Zhao et al., 2020), evidential models for regression (Amini et al., 2020, Pandey & Yu, 2023), adversarial robustness (Kopetzki et al., 2021) and calibration (Tomani & Buettner, 2021). Most existing EDL approaches typically incorporate evidential loss along with evidence regularization to guide the uncertainty behavior (Pandey & Yu, 2022, Shi et al., 2020) of the evidence. In this work, we focus on EDL loss for segmentation and DST for multi-evidence aggregation. It is a promising way to tackle the problem of segmentation map bias and semantic overlap conflict, enabling more effective image synthesis involving multiple objects.

## 5 CONCLUSION

In this paper, we formulate the problems of segmentation map bias and semantic overlap conflict in performing the consistency constraint between text prompts and image contents, and propose EviDiff, an end-to-end T2I diffusion model fine-tuning strategy equipped with consistency constraint between text prompts and image contents. We identify the causes of these two problems and propose two key components to explicitly address them. The pixel evidence loss judges the reliability of consistency supervision through pixel-level uncertainty metric. Besides, we introduce the token conflict loss to address the semantic overlap conflict among objects in the image. Despite achieving superior generation results, EviDiff shortcomings in attribute binding task. In future work, we will continue to improve this framework by considering applying constraints on color, shape, texture, etc.

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

## A  PRELIMINARY

**Stable Diffusion Model**. This work focuses on the Stable Diffusion Model (SD) [46] which functions within the latent space of an autoencoder. To be specific, an image $x_0$ is encoded to a latent code $z_0 = \mathcal{E}(x_0)$ using a variational autoencoder (VAE) [22]. Subsequently, a normally distributed noise $\epsilon$ is added to the original latent code $z_0$ with a variable extent based on a timestep $t$ sampling from $\{1, ..., T\}$. During denoising, the denoising function $\epsilon_\theta$ parameterized by a UNet [47] backbone is trained to predict the noise added to $z_0$ with the text prompt $y$ and the current latent $z_t$ as the input. The text prompt $y$ is first encoded to the text embedding $c = f_{\text{CLIP}}(y)$ through a pre-trained CLIP [42] text encoder $f_{\text{CLIP}}$. The following shows the denoising objective.

$$\mathcal{L}_{\text{LDM}} = \mathbb{E}_{z_0,t,c,\epsilon\sim\mathcal{N}(0,I)}\left[\|\epsilon - \epsilon_\theta\left(z_t, t, c\right)\|^2\right] \tag{13}$$

During inference, a latent variable $z_T$ is sampled from the standard Gaussian distribution $\mathcal{N}(0, 1)$, and then iteratively uses $\epsilon_\theta$ to estimate the noise and compute the next latent sample, thus deriving a refined latent representation $z_0$.

**Cross-attention layer**. Text image interaction in SD is achieved through the cross-attention layer, facilitating text condition guidance. Specifically, for each cross-attention layer, linear projections are employed to extract the key $K$ and value $V$ from $c$, while the query $Q$ is projected by the intermediate features of UNet. The cross-attention map $A^{(k)} \in \mathbb{R}^{h \times w \times l}$ is computed by:

$$A^{(k)} = \text{Softmax}(\frac{Q^{(k)}(K^{(k)})^T}{\sqrt{d}}) \tag{14}$$

where $k$ is the index of head. $h$ and $w$ are the resolution of the latent code. $l$ is the token length of the text embedding, and d is the feature dimension. $A_{i,j}^n$ is the attention score assigned to $n$-th text token for the $(x, y)$-th spatial patch of the intermediate feature map.

## B  PROBLEM DERIVATION

### B.1  SEGMENTATION MAP BIAS

Assuming the cross-attention map is $A_{i,j} \in [0, 1]$, it represents the probability that pixel $(i, j)$ belongs to the foreground. The true segmentation map is $M_{i,j}^* \in \{0, 1\}$. The segmentation map actually provided for model training is $M_{i,j} \in \{0, 1\}$, which is biased, defined as:

$$\varepsilon_{i,j} := M_{i,j} - M_{i,j}^* \in \{-1, 0, +1\} \tag{15}$$

The model fits the segmentation map $M_{i,j}$ with the cross-attention map $A_{i,j}$. The loss is defined as follows:

$$\mathcal{L} = -\sum_{i,j}[M_{i,j} \cdot \log A_{i,j} + (1 - M_{i,j}) \cdot \log(1 - A_{i,j})] \tag{16}$$

Bring in $M_{i,j} = M_{i,j}^* + \varepsilon_{i,j}$:

$$\mathcal{L} = -\sum_{i,j}\left[(M_{i,j}^* + \varepsilon_{i,j})\log A_{i,j} + (1 - M_{i,j}^* - \varepsilon_{i,j})\log(1 - A_{i,j})\right] \tag{17}$$

Unfold it as:

$$\mathcal{L} = -\underbrace{\sum_{i,j} \left[ M_{i,j}^* \log A_{i,j} + (1 - M_{i,j}^*) \log(1 - A_{i,j}) \right]}_{\text{Ideal supervision objective}} - \underbrace{\sum_{i,j} \left[ \varepsilon_{i,j} \log \frac{A_{i,j}}{1 - A_{i,j}} \right]}_{\text{Bias term}} \tag{18}$$

For the Bias term:

$$\mathcal{L}_{\text{bias}} := -\sum_{i,j} \varepsilon_{i,j} \log \frac{A_{i,j}}{1 - A_{i,j}} \tag{19}$$

It represents the perturbation of the biased segmentation map for model learning. We analyze different situations as follows:

*Situation 1: False Positive.* That is, $M_{i,j}^* = 0$, $M_{i,j} = 1$, then $\varepsilon_{i,j} = 1$. The bias term becomes: $-\log \frac{A_{i,j}}{1 - A_{i,j}}$. When $A_{i,j} < 0.5$, the gradient is positive, and the decrease of loss drives $A_{i,j}$ to increase. Consequently, the model is encouraged to improve the response of the error area, incorrectly focusing on foreground area.

*Situation 2: False Negative.* That is, $M_{i,j}^* = 1$, $M_{i,j} = 0$, then $\varepsilon_{i,j} = -1$. The bias term becomes: $\log \frac{A_{i,j}}{1 - A_{i,j}}$. When $A_{i,j} > 0.5$, the gradient is positive, and the decrease of loss drives $A_{i,j}$ to decrease. Consequently, the model suppresses real foreground response, ignoring the real foreground area.

Gradient Analysis: For $A_{i,j}$, its gradient is:

$$\frac{\partial \mathcal{L}}{\partial A_{i,j}} = \frac{A_{i,j} - M_{i,j}}{A_{i,j}(1 - A_{i,j})} \tag{20}$$

In the ideal situation, it should be:

$$\frac{\partial \mathcal{L}_{\text{ideal}}}{\partial A_{i,j}} = \frac{A_{i,j} - M_{i,j}^*}{A_{i,j}(1 - A_{i,j})} \tag{21}$$

Therefore, in actual training, the direction of model optimization is:

$$\frac{\partial \mathcal{L}}{\partial A_{i,j}} = \frac{A_{i,j} - M_{i,j}^* - \varepsilon_{i,j}}{A_{i,j}(1 - A_{i,j})} = \frac{\partial \mathcal{L}_{\text{ideal}}}{\partial A_{i,j}} - \frac{\varepsilon_{i,j}}{A_{i,j}(1 - A_{i,j})} \tag{22}$$

The bias term term $\frac{\varepsilon_{i,j}}{A_{i,j}(1 - A_{i,j})}$ explicitly change the direction of the gradient, which makes the biased pixel dominate the optimization direction of the model, causing the model to fit $M^* + \varepsilon$ instead of the true segmentation map $M^*$.

## B.2 SEMANTIC OVERLAP CONFLICT

Assuming the text prompt contains $N$ nouns $\{w_1, \ldots, w_N\}$, each noun $w_n$ has a corresponding true segmentation map $M^n \in \{0, 1\}$. The model predicts the cross-attention map $A_{i,j}^n \in [0, 1]$ for each $w_n$, defined as:

$$A_{i,j}^n = \sigma(\phi(z_{i,j}, e(w_n))) \quad \text{for each pixel } (i, j) \tag{23}$$

where $z_{i,j} \in \mathbb{R}^d$ is the visual feature of pixel $(i, j)$, $e(w_n) \in \mathbb{R}^d$ is the noun embedding, $\phi(\cdot)$ is the cross-attention fusion function, and $\sigma$ is the activation function.

Let the target areas of two nouns $w_1, w_2$ overlap, i.e., there exists a pixel $(i, j)$ such that:

$$M_{i,j}^1 = 1, \quad M_{i,j}^2 = 1 \tag{24}$$

We investigate whether the gradient direction of the shared image feature $z_{i,j}$ at the pixel $(i, j)$ is consistent, that is

$$\nabla_{z_{i,j}} \mathcal{L}_1 \quad \text{vs} \quad \nabla_{z_{i,j}} \mathcal{L}_2 \tag{25}$$

If they have significant differences, it indicates that there is a optimization conflict during the training process.

For noun $w_1$, Let:

$$a_1 = \phi(z_{i,j}, e(w_1)), \quad A_{i,j}^1 = \sigma(a_1) \tag{26}$$

Then:

$$\frac{\partial \mathcal{L}_1}{\partial z_{i,j}} = \frac{\partial \mathcal{L}_1}{\partial A_{i,j}^1} \cdot \frac{dA_{i,j}^1}{da_1} \cdot \frac{\partial a_1}{\partial z_{i,j}} \tag{27}$$

Expand item by item:

$$\frac{\partial \mathcal{L}_1}{\partial A_{i,j}^1} = -\frac{M_{i,j}^1}{A_{i,j}^1} + \frac{1 - M_{i,j}^1}{1 - A_{i,j}^1} \tag{28}$$

$$\frac{dA_{i,j}^1}{da_1} = A_{i,j}^1(1 - A_{i,j}^1) \tag{29}$$

$$\text{if } \phi(z_{i,j}, e(w_1)) = z_{i,j}^\top e(w_1) \quad \text{then } \frac{\partial a_1}{\partial z_{i,j}} = e(w_1) \tag{30}$$

Based on the above analysis:

$$\nabla_{z_{i,j}} \mathcal{L}_1 = \left( -\frac{M_{i,j}^1}{A_{i,j}^1} + \frac{1 - M_{i,j}^1}{1 - A_{i,j}^1} \right) \cdot A_{i,j}^1(1 - A_{i,j}^1) \cdot e(w_1) \tag{31}$$

By analogy:

$$\nabla_{z_{i,j}} \mathcal{L}_2 = \left( -\frac{M_{i,j}^2}{A_{i,j}^2} + \frac{1 - M_{i,j}^2}{1 - A_{i,j}^2} \right) \cdot A_{i,j}^2(1 - A_{i,j}^2) \cdot e(w_2) \tag{32}$$

Define the overlapping area between $w_1$ and $w_2$ as:

$$\mathcal{O}_{i,j} = \{(i,j) \in [H] \times [W] \mid M_{i,j}^1 = 1 \text{and} M_{i,j}^2 = 1\} \tag{33}$$

For the pixel $(i,j) \in \mathcal{O}_{i,j}$,

$$\nabla_{z_{i,j}} \mathcal{L}_1 \propto (-\frac{1}{A_{i,j}^1}) \cdot A_{i,j}^1(1 - A_{i,j}^1) \cdot e(w_1) = -(1 - A_{i,j}^1) \cdot e(w_1) \tag{34}$$

By analogy:

$$\nabla_{z_{i,j}} \mathcal{L}_2 \propto -(1 - A_{i,j}^2) \cdot e(w_2) \tag{35}$$

Finally, we obtained:

$$\nabla_{z_{i,j}} \mathcal{L}_{\text{total}} = \nabla_{z_{i,j}} \mathcal{L}_1 + \nabla_{z_{i,j}} \mathcal{L}_2 = -[(1 - A_{i,j}^1)e(w_1) + (1 - A_{i,j}^2)e(w_2)] \tag{36}$$

This is a weighted sum of two nouns $w_1, w_2$. If $e(w_1)$ and $e(w_2)$ has semantic overlap, then there will be conflict in the overall gradient direction during optimization.

Specifically, when:

$$\langle e(w_1), e(w_2) \rangle < 0 \tag{37}$$

The two gradient directions cancel each other out, ultimately leading to:

$$\|\nabla_{z_{i,j}} \mathcal{L}_{\text{total}}\| < \min\{\|\nabla_{z_{i,j}} \mathcal{L}_1\|, \|\nabla_{z_{i,j}} \mathcal{L}_2\|\} \tag{38}$$

It causes the gradient conflict.

## C  DETAILS OF EVALUATION METRICS

We measure the following metrics: **1) VISOR** [13] can assess how accurately the spatial relationship described in text is generated in the image. The VISOR benchmark generates all unique pairwise combinations of 80 COCO object categories. Each pair (A, B) is converted into a text prompt following a template "¡A¿¡R¿¡B¿," where R represents an arbitrary spatial relationship. For example, one such prompt could be "a cat to the right of a table." Finally, 31600 text prompts were constructed and used as conditions to generate 31600 images. Similarity, the open-vocabulary detector [36] is used to detect the presence of each category in each generated image and the Object Accuracy (OA) is employed to evaluate how successfully instances are generated from each of the two categories. **2) MULTIGEN** [55] is used to evaluate the ability of the model in combining instances of multiple categories. Specifically, given a set of $N$ distinct instance categories, five categories (e.g., A, B, C, D, and E) are randomly selected and formatted into a sentence (e.g., "A photo of A, B, C, D, and E"). This sentence serves as the conditional input for the T2I diffusion model to generate the corresponding image. Subsequently, a robust open-vocabulary detector [36] is employed to

assess whether the specified categories are accurately represented in the generated image. 1,000 text prompts are built by randomly sampling 80 categories of COCO instances [28], which are used as inputs for the multi-category instance combinations. To avoid inference variance, each prompt is used to generate 10 rounds of images, resulting in a total of $10 \times 1000$ images. For each generated image, the open-vocabulary detector is used to determine how many different categories of objects. Based on detection results, MG2-5 metrics are defined through the overall success rates of generating 2-5 specified categories from 5 categories. **3) Fréchet Inception Distance (FID)** [15] is used to assess the quality of the generated images from two datasets: i) 25014 image-caption pairs sampled from the COCO instance validation set; ii) 5000 image-caption pairs sampled from the Flickr30K instance validation set [39]. **4) CLIP Score** [25] is utilized to evaluate realism, which reflects the degree of match between generated images and text prompts. The evaluation datasets is the same as FID. **5) Efficiency** is used to compare the inference-time (i.e., the time required to generate an image using the trained model) of our EviDiff with other baselines. **5) T2I-CompBench** [16] comprises 6,000 compositional text prompts evaluating 4 categories (attribute binding, object relationships, Numeracy, and complex compositions) and 7 sub-categories (color binding, shape binding, texture binding, spatial relationships, non-spatial relationships, numeracy and complex compositions).

## D  DETAILS OF BASELINES

**(1) Stable Diffusion** [46], which improves traditional diffusion models by reducing the high computational cost. **(2) Composable Diffusion** [30], which composes a set of diffusion models, with each of them modeling a certain component of the image. **(3) Layout Guidance Diffusion** [6], which proposes backward guidance to bias the cross-attention via energy minimization for desired layout matching. **(4) Structured Diffusion** [10], which employs consistency trees or scene graphs to split the prompt into several noun phrases and manipulates the cross-attention layer for image generation. **(5) Attend-and-Excite** [4], which refines the cross-attention units to focus on all subject tokens in the text prompt, encouraging the model to generate all subjects described in the text prompt. **(6) TokenCompose** [55], which conducts consistency constraint between text prompts and the image contents for multi-category instance composition.

## E  DETAILS OF FINETUNING AND INFERENCING

In **Algorithm 1**, We provide a detailed finetuning process for EviDiff, which addresses the segmentation map bias and semantic overlap conflict during the consistency constraint between the text prompts and the image contents, by pixel-level uncertainty estimation and token-level conflict optimization. The pseudocode for the denoising process of EviDiff is as follows.

---

**Algorithm 1** Finetuning for EviDiff

---

**Require:** A text prompt $y$, a clear image $x_0$, and a pretrained stable diffusion model $\epsilon_\theta$.

1: $t \sim \text{Uniform}(\{1, \ldots, T\})$
2: $\epsilon \sim \mathcal{N}(\mathbf{0}, \mathbf{I})$
3: $c = f_{\text{CLIP}}(y)$
4: $z_t = \sqrt{\bar{\alpha}_t} f_{VAE}(x_0) + \sqrt{1 - \bar{\alpha}_t} \epsilon$
5: **for** step $= 1, \ldots, S$ **do**
6: $\quad \mathcal{L}_{\text{LDM}} = \mathbb{E}_{z_0, t, c, \epsilon \sim \mathcal{N}(0, I)} \left[ \|\epsilon - \epsilon_\theta (z_t, t, c)\|^2 \right]$
7: $\quad$ **for** layer $= \text{mid8}, \text{up16}, \text{up32}, \text{up64}$ **do**
8: $\quad\quad$ get the noun token's cross-attention maps $[A_{(1)}, \ldots, A_{(N)}]$ from each layer
9: $\quad\quad$ get the noun token's segmentation maps $[M_{(1)}, \ldots, M_{(N)}]$ from SAM
10: $\quad\quad$ **for** $(A_{(n)}, M_{(n)}) = (A_{(1)}, M_{(1)}), \ldots, (A_{(N)}, M_{(N)})$ **do**
11: $\quad\quad\quad$ compute $\mathcal{L}_{\text{PixEvi}}^{(n)}(A_{(n)}, M_{(n)})$ from Eq. 7
12: $\quad\quad\quad$ compute $\mathcal{L}_{\text{TokCon}}^{(n)}(A_{(n)}, A_{(n+1)})$ from Eq. 12
13: $\quad\quad$ **end for**
14: $\quad$ **end for**
15: $\quad$ get $\mathcal{L}_{\text{EviDiff}}$ from Eq. **??**
16: **end for**

---

In **Algorithm 2**, We provide a detailed denoising process for EviDiff, which can be applied directly to the existing training pipeline of T2I diffusion models.

---

**Algorithm 2** Inferencing for EviDiff

    **Input:** A text prompt $y$ and the trained EviDiff
    **Output:** A clear latent $z_0$
1: $z_T \sim \mathcal{N}(\mathbf{0}, \mathbf{I})$
2: $c = f_{\text{CLIP}}(y)$
3: **for** $t = T, \ldots, 1$ **do**
4:     $z_{t-1} = \frac{1}{\sqrt{\alpha_t}} \left( z_t - \frac{1-\alpha_t}{\sqrt{1-\bar{\alpha}_t}} \epsilon_\theta(z_t, t, c) \right)$
5: **end for**
6: **return** $z_0$

---

## F  HYPERPARAMETER ABLATION STUDY

Table 5: Results with different values of $\lambda_1$.

| $\lambda_1$ | OA↑ | FID(C)↑ | FID(F)↑ | CLIP(C)↑ | FID(F)↑ |
|---|---|---|---|---|---|
| 5e-4 | 0.7109 | 20.11 | 55.26 | 0.3229 | 0.3330 |
| 5e-5 | 0.7116 | 20.09 | 55.12 | 0.3241 | 0.3348 |
| 5e-6 | 0.7110 | 20.13 | 55.24 | 0.3237 | 0.3336 |

Table 6: Results with different values of $\lambda_2$.

| $\lambda_2$ | OA↑ | FID(C)↑ | FID(F)↑ | CLIP(C)↑ | FID(F)↑ |
|---|---|---|---|---|---|
| 5e-6 | 0.7105 | 20.17 | 55.33 | 0.3230 | 0.3328 |
| 5e-7 | 0.7116 | 20.09 | 55.12 | 0.3241 | 0.3348 |
| 5e-8 | 0.7097 | 20.14 | 55.15 | 0.3234 | 0.3343 |

Table 7: Results with different values of $\eta_1$.

| $\eta_1$ | OA↑ | FID(C)↑ | FID(F)↑ | CLIP(C)↑ | FID(F)↑ |
|---|---|---|---|---|---|
| 1e-3 | 0.7109 | 20.31 | 55.54 | 0.3226 | 0.3324 |
| 1e-4 | 0.7116 | 20.09 | 55.12 | 0.3241 | 0.3348 |
| 1e-5 | 0.7102 | 20.28 | 55.32 | 0.3230 | 0.3315 |

Table 8: Results with different values of $\eta_2$.

| $\eta_2$ | OA↑ | FID(C)↑ | FID(F)↑ | CLIP(C)↑ | FID(F)↑ |
|---|---|---|---|---|---|
| 1e-3 | 0.7103 | 20.46 | 55.44 | 0.3213 | 0.3320 |
| 1e-4 | 0.7116 | 20.09 | 55.12 | 0.3241 | 0.3348 |
| 1e-5 | 0.7088 | 20.38 | 55.65 | 0.3219 | 0.3314 |

## G   GENERATION RESULTS FINETUNED BASED ON STABLE DIFFUSION V2.1 AND SD V3 MEDIUM

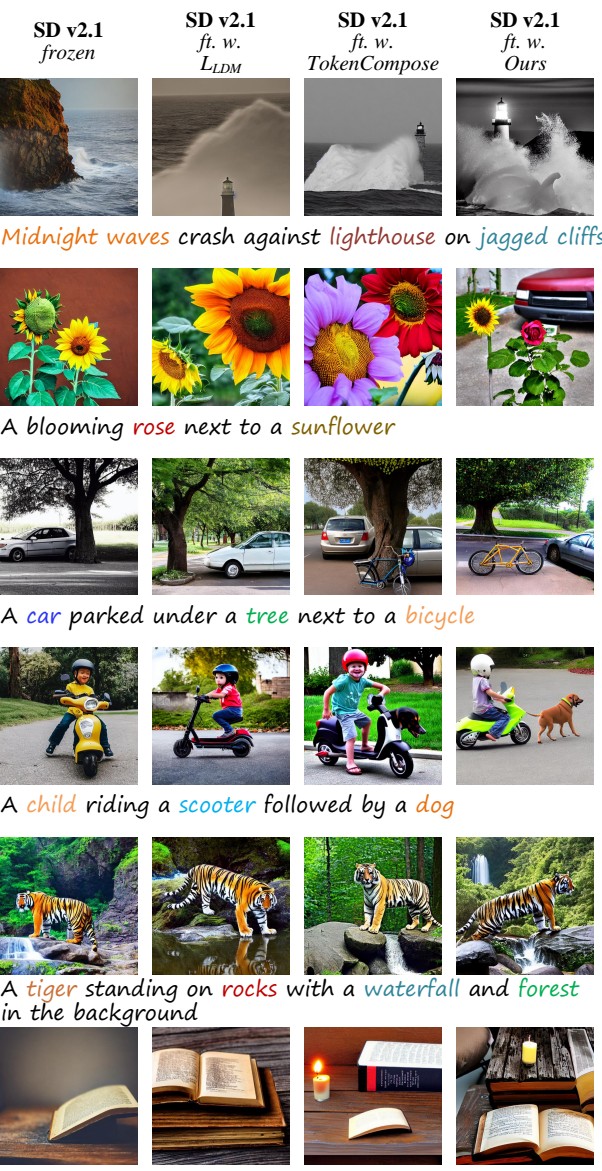

Figure 6: Qualitative comparison of finetuning methods based on Stable Diffusion v2.1.

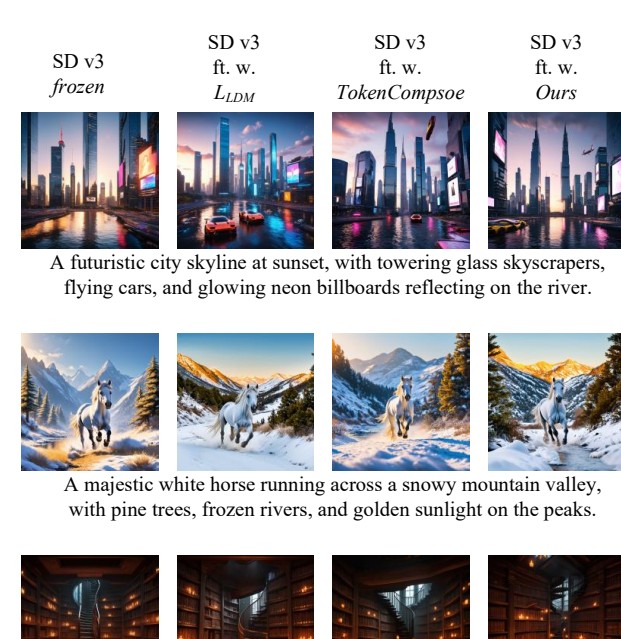

Figure 7: Qualitative comparison of finetuning methods based on SD v3 Medium.

## H  THE USE OF LLM

In this work, we employ a large language model (LLM) solely for language polishing of the manuscript.

