# OpenReview forum: "EviDiff: Learning Object-wise Consistency for Text-to-Image Diffusion"
_ICLR.cc/2026/Conference — ICLR 2026 Conference Withdrawn Submission_

### Official Review · Reviewer_t62f · 2025-10-30

**Soundness:** 2
**Presentation:** 2
**Contribution:** 2
**Rating:** 4
**Confidence:** 3

**Summary:**

* The paper introduces EviDiff, a novel T2I model trained with evidential learning for improved object consistency.
  * The authors propose a pixel evidence loss to manage overconfidence in unreliable segmentation labels by simultaneously considering pixel-level classification prediction and its associated uncertainty estimation.
  * The authors design a token conflict loss is designed to weaken semantic overlap conflict by using a conflict factor, measured via the DST combination rule, directly as the optimization objective.
* The authors proved the effectiveness of the proposed EviDiff method by showing it outperforms multiple baseline models and previously proposed training approaches.

**Strengths:**

* The proposed EviDiff method is a novel of Evidential Deep Learning (EDL) and Dempster-Shafer Theory (DST) to a T2I diffusion model, which is a highly original approach for handling compositional conflicts.
* The proposed pixel evidence loss targes the issue of segmentation map bias by managing overconfidence in unreliable labels through uncertainty estimation.
* The token conflict loss optimizes against the semantic overlap conflict to tackle this common problem in multi-object compositional generation.
* Empirical Results show that EviDiff outperform multiple baselines and previous works.
* EviDiff can be extended to existing T2I diffusion model training pipelines without requiring additional inference-time manipulations.

**Weaknesses:**

* **Empirical Improvements seem trivial.** Improvements on most quantitative metrics seem trivial (e.g. MG3, MG4, CLIP(C), FID(F), FID(C) metrics in Table 2) and sometimes falling behind previous methods (e.g. MG2, MG5 in Table 2). It is hard to justify that the proposed method brings significant enough contributions.
* **Limited Model Architecture Studied**. Experiments were only conducted on the older generation of T2I models like SD 1.4, SD 2.1, and SD 3. Newer models like SD3.5, FLUX, and Qwen-Image are not explored.
  * One related point is that, with the stronger generation of T2I models coming out (especially closed-source ones like gpt-image-1 and nano-banana), the object consistency setting studied in this paper might have been fully resolved for newer models. Consistency in more complex generation settings (e.g. multiple objects with different spatial and appearance attributes) might still be challenging, but it is out of the scope of this study, too.

**Questions:**

* How would your method perform when applied to newer models like FLUX, UniWorld and Qwen-Image?
* How well do more recent T2I models perform on the object consistency task?

---

### Official Review · Reviewer_H9Tj · 2025-10-31

**Soundness:** 3
**Presentation:** 2
**Contribution:** 2
**Rating:** 4
**Confidence:** 5

**Summary:**

The paper proposes EviDiff, a fine-tuning strategy for text-to-image (T2I) diffusion models to improve multi-object compositionality. It leverages a pixel evidence loss that models cross attention as a Dirichlet distribution to account for noisy segmentation masks, and a token conflict loss to address overlapping token attentions. Authors validated their objectives by training various stable diffusion models on COCO datasets with segmentation masks, and showed consistent gains over baselines e.g., TokenCompose for different models and different metrics.

**Strengths:**

- The objectives are well-motivated e.g., uncertainty on masks and token mask overlaps and are derived from principled methodologies. Empirical evaluation demonstrates their effectiveness on combinations of models and evaluations for instance-centric compositionality.
- Training is drop-in with the new objective and the method does not rely on additional compute during inference.

**Weaknesses:**

##

- The method mainly integrates evidential learning and overlap/conflict penalties of attention-guidance to the TokenCompose recipe, both of which are existing principles. These are useful empirically, but the contribution feels incremental. Also, the improvements with the TokCon objective appear to be marginal — it would be better to examine whether these numbers are statistically significant by performing multiple training runs and taking account the number of samples used in evaluation into account. Though, my major concern is the selection of models and evaluations being used (see below).
- Most of results are based on SD 1.4, 2.1 and some are based on SD 3 — but these are no longer representative of current T2I models. It would be great to show if these methods can be adapted to finetune current T2I and unified models (e.g., flux 1, sd 3.5; qwen image, bagel) to achieve a more competitive performance for T2I compositionality.
- Evaluation set is also outdated. It would be great to show methods’ effectiveness on modern benchmarks for compositionality e.g., T2I-CompBench++, GenEval, GenAI-Bench as they evaluate more than multi-category instance generation; by training newer models and evaluating them on newer benchmarks, the paper can better substantiate its claims beyond dated CLIP/FID-style proxies and small suites.

**Questions:**

- Are there any training instabilities when you add those objectives, and how do you tune all the scalar hyperparameters in these objectives?
- Because most of gains come from PixEvi objective, I wonder if you could do a fine-grained study on different components of this objective i.e., L_ce, L_ice and L_KL and evaluate their effectiveness individually and when combined as pairs?
- Do the methods help attribute binding beyond nouns (colors/textures/relations)? i.e., does the probability of having the correct color/texture/relation increase when conditioned on being able to generate a specific instance in multi-category instance generation?

---

### Official Review · Reviewer_fTDj · 2025-11-01

**Soundness:** 3
**Presentation:** 3
**Contribution:** 2
**Rating:** 4
**Confidence:** 5

**Summary:**

This paper proposes **EviDiff**, a training framework designed to improve object-wise consistency in text-to-image diffusion models. The authors identify two key issues in prior token-level supervision approaches (e.g., TokenCompose): (1) segmentation mask noise that produces unreliable pixel-token correspondence, and (2) inter-token conflicts when multiple objects overlap semantically or spatially. To address these, EviDiff introduces two complementary losses: a *pixel evidence loss* that interprets attention activations as Dirichlet evidences to down-weight uncertain supervision, and a *token conflict loss* based on Dempster–Shafer Theory (DST) to penalize conflicts between overlapping object regions. The method is applied as a fine-tuning layer to Stable Diffusion 1.4, 2.1, and 3-Medium, improving multi-object composition benchmarks such as VISOR and MULTIGEN, while maintaining comparable inference speed.

**Strengths:**

- The paper provides a clear motivation for addressing segmentation noise and token-level conflict in compositional generation.
- The pixel evidence loss elegantly integrates uncertainty modeling from evidential deep learning, allowing the model to handle unreliable labels more robustly.
- The token conflict loss formalizes a DST-based measure to explicitly reduce conflicts between overlapping objects, a conceptually sound and novel approach.
- Experimental results across SD variants demonstrate consistent improvement in compositional accuracy and FID/CLIP scores, without increasing inference cost.
- The approach is lightweight and easily integrates into existing diffusion model training pipelines.

**Weaknesses:**

- Algorithm boxes contain a dangling reference (“get LEviDiff from Eq. ??”, Algorithm 1 in Appendix E), suggesting a missing equation label in the main text.
- The evidential modeling details are under-specified: the class count K for the Dirichlet (foreground/background) and how multi-head/multi-layer attention maps are combined before evidence estimation could be clearer; the choice of layers (mid8, up16, up32, up64) is given but not justified.
- Reliance on auto-generated segmentation without any human verification raises concerns about confirmation bias in both training and evaluation; the same detector family is used for evaluation (open-vocabulary detector), which may correlate with the supervision signal.
- The overall writing style, structure, and even visual presentation (e.g., layout, typography, figure design) are very similar to TokenCompose, to the extent that it is difficult to determine whether this is an independent follow-up or an extension.

**Questions:**

Please refer to the weakness section

---

### Official Review · Reviewer_LNzA · 2025-11-01

**Soundness:** 2
**Presentation:** 2
**Contribution:** 2
**Rating:** 4
**Confidence:** 3

**Summary:**

The authors observe that current text-to-image (T2I) diffusion models struggle to reliably compose multiple objects when guided by a full text prompt, due to weak object-wise consistency between textual tokens and image content. Prior token-supervised methods help by aligning each noun token with a segmentation map, but suffer from segmentation map bias (inaccurate masks) and semantic overlap conflicts (when object regions overlap). To address these, EviDiff introduces two key losses: a pixel evidence loss which uses evidential deep learning to model the prediction uncertainty of each pixel (thereby reducing reliance on unreliable segmentation maps), and a token conflict loss based on Dempster-Shafer theory to quantify and suppress conflict between overlapping object token attentions. The approach can be applied as a fine-tuning add-on to existing diffusion models, yields improved multi-object composition, photorealism, and inference efficiency, and outperforms prior methods in various benchmarks.

**Strengths:**

1. EviDiff clearly identifies two concrete failure modes in text-to-image diffusion — segmentation bias and semantic overlap conflict — and proposes principled solutions using evidential deep learning and Dempster-Shafer theory. The combination of pixel evidence loss and token conflict loss is conceptually sound, theoretically grounded, and effectively targets weaknesses in existing token-supervised methods.

2. The paper demonstrates consistent improvements in multi-object compositionality, object-wise consistency, across several benchmarks and base diffusion models. Its plug-and-play fine-tuning nature allows integration with different pretrained generators without architectural modification, enhancing its practical relevance and reproducibility.

**Weaknesses:**

1. While EviDiff introduces new loss formulations, its core idea—enhancing object-wise consistency through token-level supervision—largely follows the same motivation as TokenCompose. The proposed pixel evidence and token conflict losses refine rather than redefine this framework, leading to improvements that appear incremental rather than conceptually groundbreaking.

2. The paper lacks a deeper theoretical justification or interpretability study explaining why evidential learning and Dempster-Shafer conflict modeling yield better compositional alignment. The improvements are shown empirically, but without clear insights into how uncertainty estimation interacts with diffusion attention dynamics. Additionally, an intuitive example/visualization could help clarify the motivation for introducing uncertainty quantification.

3. Experiments focus primarily on controlled multi-object settings. It remains unclear how EviDiff performs on broader compositional prompts involving spatial relationships and verbs.

**Questions:**

1. Since SAM is not entirely reliable, have the authors considered using a more advanced segmentation model, such as SAM 2, to improve the accuracy of object masks and reduce segmentation bias? Of course, SAM2 + EviDiff may bring additional improvement. But it is interesting to know if the proposed method can still be effective when the segmentation model's capability has been improved a lot.

2. Could the authors clarify the actual computational difference between the proposed intra loss (Eq. 10) and the L_token loss used in TokenCompose? A more detailed comparison would help readers understand the novelty and efficiency of the proposed formulation.

3. Is there any difference in the attention layers used or how the proposed attention mechanism behaves across different architectural designs, such as diffusion models based on U-Net (e.g., SDv2) versus Diffusion Transformers (e.g., SDv3)? It would be helpful to discuss whether architectural differences affect attention consistency or loss of effectiveness.

---

### Note · Authors · 2025-11-12

I have read and agree with the venue's withdrawal policy on behalf of myself and my co-authors.